# Genomics, Origin and Selection Signals of Loudi Cattle in Central Hunan

**DOI:** 10.3390/biology11121775

**Published:** 2022-12-07

**Authors:** Liangliang Jin, Baizhong Zhang, Jing Luo, Jianbo Li, Juyong Liang, Wanghe Wu, Yongzhong Xie, Fuqiang Li, Chuzhao Lei, Kangle Yi

**Affiliations:** 1Hunan Institute of Animal and Veterinary Science, Changsha 410130, China; 2Key Laboratory of Animal Genetics, Breeding and Reproduction of Shaanxi Province, College of Animal Science and Technology, Northwest A&F University, Yangling, Xianyang 712100, China; 3Qinghai Academy of Animal Science and Veterinary Medicine, Qinghai University, Xining 810016, China; 4Animal Husbandry and Fishery Affairs Center of Lianyuan City, Loudi 417100, China; 5Hunan Tianhua Industrial Co., Ltd., Lianyuan 417126, China

**Keywords:** whole-genome resequencing, genetic diversity, admixture events, hybrid cattle

## Abstract

**Simple Summary:**

Similar cattle breeds are difficult to distinguish based on appearance alone, but their price and breeding options vary greatly. In this study, we examined a high level of genomic diversity in Loudi cattle from central Hunan using whole-genome resequencing data. We speculated the origin of Loudi cattle and detected the positive selection signal of its genome by comparison with carefully selected cattle breeds from around the world. These findings will serve as the basis for upcoming conservation and breeding initiatives.

**Abstract:**

Due to the geographical, cultural and environmental variability in Xiangxi, China, distinctive indigenous cattle populations have formed. Among them, Loudi cattle and Xiangxi cattle are the local cattle in Hunan, and the environment in Loudi is relatively more enclosed and humid than that in Xiangxi. To study the genome and origin of Loudi cattle in hot and humid environments, 29 individuals were collected and sequenced by whole-genome resequencing. In addition, genomic data were obtained from public databases for 96 individuals representing different cattle breeds worldwide, including 23 Xiangxi cattle from western Hunan. Genetic analysis indicated that the genetic diversity of Loudi cattle was close to that of Chinese cattle and higher than that of other breeds. Population structure and ancestral origin analysis indicated the relationship between Loudi cattle and other breeds. Loudi has four distinctive seasons, with a stereoscopic climate and extremely rich water resources. Selective sweep analysis revealed candidate genes and pathways associated with environmental adaptation and homeostasis. Our findings provide a valuable source of information on the genetic diversity of Loudi cattle and ideas for population conservation and genome-associated breeding of local cattle in today’s extreme climate environment.

## 1. Introduction

Domestic cattle provide humans with important labor and rich material resources, supporting half of China’s farming civilization. Modern domestic cattle generally consist of *Bos taurus* and *Bos indicus* lineages [1]. *Bos taurus* mainly originated from cattle in Europe and has advantages in beef and milk production after continuous artificial breeding, while *Bos indicus* originated from cattle in South Asia and is able to tolerate heat and crude feed [2,3]. Geographically, domestic cattle may be loosely classified into five groups: European taurine, Eurasian taurine, East Asian taurine, Chinese indicine and Indian indicine [4]. With the flow of the long history, some local cattle breeds were subject to introgression of other breeds [5]. In addition, to better adapt to the complicated environment, the continuous accumulation of genetic variation in the genome has created today’s rich resources of local cattle breeds [6].

Since ancient times, Hunan Province has evolved an agricultural culture, prosperous animal husbandry and rich cattle resources. Loudi cattle are one of the precious livestock genetic resources of Hunan cattle, originating in Loudi City, the geographic and geometric center of Hunan Province. The local ancients (approximately 500–550 AD) believed that cattle existed before humans and held an annual cattle King Festival. This showed that cattle were highly valued and widely used in production work at that time, and it also indicated that the Loudi area and even the Chinese civilization had changed from a fishing and hunting culture to a farming civilization [7].

Numerous studies on the genetic diversity and lineage origins of local cattle breeds are becoming more prevalent as the use and price of whole-genome sequencing increase. [8,9]. However, there are no previous studies on Loudi cattle from central Hunan, only research work on Xiangxi cattle from western Hunan has been found [10]. It is worth noting that Hunan is dominated by mountains and hills, especially Loudi, which is surrounded by mountains, belonged to the Miao Yao minority in ancient times and rarely communicates with the outside world. Therefore, compared with Xiangxi cattle, which have formed a relatively mature reserved area and industrial chain, Loudi does not even have a large breeding cattle farm, and most of the Loudi cattle are bred by scattered farmers.

Here, we performed whole-genome sequencing of 29 individuals of Loudi cattle to explore the genetic diversity and population genetic structure of the autosomal genome and to analyze the origin, introgression and admixture of Loudi cattle.

## 2. Materials and Methods

### 2.1. Ethics Statement

The study was approved by the Institutional Animal Care and Use Committee of Northwest A&F University. All the applicable institutional and national guidelines for the care and welfare of animals have been strictly followed for the tissue sampling procedures.

### 2.2. Sample Collection and Sequencing

Due to the lack of local cattle reservation areas in Loudi, we selected 29 unrelated cattle, including 26 cows and 3 bulls, from multiple farms in the Loudi area of Hunan Province, China. We collected ear tissue samples and extracted DNA using standard methods [11]. After the examinations, each eligible sample had paired-end libraries created with an average read length of 150 bp and an insert size of 500 bp. All libraries were sequenced on an Illumina NovaSeq 6000 platform. We finally obtained 1077.61 Gb of whole-genome sequencing data. In addition, resequencing data of 96 individuals were downloaded from public databases, resulting in a total of 125 samples for this study.

### 2.3. Read Mapping and Variant Calling

Trimmomatic v0.38 [12] was used to filter paired-end sequence data. BWA-MEM [13] was used to align the clean reads to the *Bos taurus* reference assembly ARS-UCD1.2. In the mapping process, a BAM file index was built using samtools [14], following which the BAM files were sorted and potential duplicate reads were deleted using picard tools (http://broadinstitute.github.io/picard, accessed on 23 June 2020). After mapping, the “Haplotype Caller”, “Genotype GVCFs” and “Select Variants” modules of the Genome analysis toolkit (GATK, version 3.8-1-0-gf15c1c3ef) [15] were used to call the SNPs. The mapping and filtering parameters were as shown previously [8]. SNPs were functionally annotated using ANNOVAR [16].

### 2.4. Genetic Diversity Analysis

After linkage disequilibrium (LD) pruning of SNPs, the runs of homozygosity (ROH) were calculated with the PLINK program [17] using the following parameters: (a) 100 consecutive homozygous SNPs; (b) minimum 50 SNPs per window; (c) 500 kb homozygous length; (d) minimum density of SNPs was 1 per 50 kb; (e) the window overlap ratio was 0.05; (f) 1 heterozygous and 2 missed calls per window. The ROHs were visualized in three different size classes (0.5–1 Mb, 1–2 Mb, 2–4 Mb), which represent the inbreeding of ancient, historical and recent times, respectively [18,19,20]. We estimated the inbreeding coefficient (Fhom) based on SNP heterozygosity [21]. The nucleotide diversity of each breed was investigated by VCFtools [22] with a 50 kb non-overlapping window. To evaluate LD with physical distances between SNPs within haplotype blocks in different cattle groups, LD decay was computed using PopLDdecay software [23]. In addition, ROH island frequencies in Loudi cattle were calculated by dividing the number of consensus samples by total samples, and the regions having a frequency of at least 20% were identified as hot spot ROH islands.

### 2.5. Population Structure Analysis

Three methods were used to estimate population structure based on filtered VCF files. Principal component analysis (PCA) was calculated using the EIGENSOFT software v5.0 [24]. Population structure analysis was carried out by ADMIXTURE v1.3.0 [25]. Cross-validation was performed to calculate the CV error and choose the optimum K values. The neighbor-joining tree was constructed using MEGA v10.2.6 [26,27]. Lastly, the tree was visualized by iTOL [28].

### 2.6. Admixture Event Ancestry Inference

We explored the full ancestral topology of an admixture graph space using Admixtools2 (see documentation at https://uqrmaie1.github.io/admixtools, accessed on 23 August 2022). This is more automated and is becoming a common approach within the field. The module ‘qpGraph’ was used to estimate f3-statistics and the topology of an admixture graph to find the best-fitting model. The output suggested the most likely relationships between the input groups.

Furthermore, we performed ancestry inference on autosomes of the Loudi cattle genome using RFmix [29], which divided each chromosome into windows and inferred local ancestry within each window by using a conditional random field parameterized by random forests trained on reference panels. Each window ancestral probability was Z-normalized, and then the cumulative distribution probability test was performed. Only windows with P values less than 0.05 were visualized and annotated.

### 2.7. Detection of Selection Signatures

The nucleotide diversity analysis (θπ) and the composite likelihood ratio test (CLR) [30] were used to detect genomic regions related to selection in Loudi cattle. We used VCFtools to obtain θπ from the VCF file. In our case, we estimated the value of π from a 50 kb window of the genome. The CLR test calculated differences in allele frequencies using SweepFinder2 [31], which implements a likelihood-based method to calculate the statistic for each site with a 20 kb grid size.

The fixation index (*F*_ST_) and cross-population extended haplotype homozygosity (XP-EHH) [32] were used to estimate differential signals between groups. *F*_ST_ analysis was performed to evaluate fixation using VCFtools. XPEHH used the long-range haplotype (LRH) test to look for alleles of high frequency with long-range linkage disequilibrium.

### 2.8. Enrichment Analyses and Visualization

The enrichment module in KOBAS intelligent version (http://bioinfo.org/kobas, accessed on 2 September 2022) [33] was used to determine which pathways and Gene Ontology (GO) terms were statistically significantly associated with the input gene list. Statistical analysis and visualization were performed using the R 4.0.5 software package (http://www.r-project.org, accessed on 18 November 2021).

### 2.9. Data Availability

Raw reads of the 29 Loudi cattle can be obtained from the National Center for Biotechnology Information (NCBI) via BioProject ID PRJNA886655. The public data were downloaded from the Sequence Read Archive (SRA, https://www.ncbi.nlm.nih.gov/sra, accessed on 20 August 2022) [34] and the Genome Sequence Archive (GSA, https://ngdc.cncb.ac.cn/gsa, accessed on 20 August 2022) [35] in the National Genomics Data Center [36]. Correspondence associated with the samples and database is shown in Appendix A.

## 3. Results

### 3.1. Genome Resequencing, SNP Detection and Diversity

A total of 125 resequencing data were selected for subsequent analysis. The average sequencing depth of the complete aligned read set of all individuals was estimated at 13.0×, with an average mapping rate of 99.21%. Among them, the average sequencing depth of Loudi cattle was 15.6×, and the average mapping rate was 99.75% (Appendix A).

We identified 184,512,060 SNPs across the whole genome of all samples. The mean number of SNPs per individual of Loudi cattle (1,351,108) was less than that of Xiangxi cattle (1,824,532) and close to that of Simmental cattle (1,234,737). In addition, the intragenic variant in Loudi cattle (32) was much smaller than that in Xiangxi cattle (446), although Loudi cattle had the largest number of individuals. Furthermore, 1.28% of the SNPs were present in exon regions, including 626,274 synonymous SNPs (Appendix A).

We investigated genomic patterns indicative of recent demographic history based on the detection of ROHs (Figure 1A). Short ROHs, which are a manifestation of ancient inbreeding, were detected in far higher numbers in European commercial cattle than in other cattle breeds. Loudi cattle exhibited a heavier ROH burden than Xiangxi cattle, suggesting a longer history of breeding. For the inbreeding coefficient based on genome heterozygosity per individual, Angus was the highest (0.71), and Loudi cattle had the greatest span (−0.16~0.30) (Figure 1B). In addition, Loudi cattle showed the fastest rate of LD decay (Figure 1C), indicating a low degree of domestication and high genetic diversity, which was verified by nucleotide diversity analysis (Figure 1D).

Five hot spot ROH islands with frequencies of 20% were identified on Loudi cattle chromosome 7 (Figure 1E). These ROH islands were annotated with 26 genes (Appendix A), including three heat-shock protein genes (*HSPA9*, *HSPC195* and *DNAJC18*), four genes (*EGR1* [37], *UBE2D2* [38], *TMEM173* [39] and *REEP2* [40]) involved in the inflammatory response and one neuregulin gene (*NRG2*).

### 3.2. Population Structure and Demography

To evaluate population stratification and admixture, we constructed a phylogenetic tree and performed PCA as well as population structure analysis of whole-genome SNPs of the eight cattle populations. The phylogenetic relationship is shown in Figure 2A. The first eigenvector distinguished *Bos taurus* and *Bos indicus* lineages. The eight populations were clustered separately into three distinct genetic groups. Loudi cattle were intermediate between Chinese cattle and Xiangxi cattle, while Xiangxi cattle were closer to taurine cattle. Similarly, the NJ tree revealed that the hybrid group showed the closest affinity for Loudi cattle and Xiangxi cattle (Figure 2B). Interestingly, the two Loudi individuals classified as the Indian cattle clade tree are both bulls. Admixture analysis (K = 2) showed that indicine cattle were separated from taurine cattle, with Hunan hybrid cattle occupying the middle (Figure 2C). When K = 3, Chinese indicine showed ancestral compositions that differed from those of Indian indicine. Loudi cattle and Xiangxi cattle had the same ancestral composition when K ranged from 2 to 4, suggesting that these had evolved from the same ancestral population. Specifically, Xiangxi cattle showed more *Bos taurus* lineage, which means a higher degree of hybridization.

### 3.3. Origin and Ancestry Inference

We took an automatic iterative approach to explore the population history of all groups. Hereford is an outgroup to all domestic cattle, with the highest allele frequency correlation among European commercial cattle (Figure 3A). The fitting results showed that some gene flow between East Asian taurine and Chinese indicine resulted in a population ancestral to Xiangxi cattle. Gene flow between Chinese cattle at different stages suggests that the *Bos indicine* lineage of Loudi cattle was not a single structure. In addition, the topological map also supported the result that the infiltration of taurine cattle was more serious in Xiangxi cattle (~24%), and the admixture event of Loudi cattle occurred much later than that of Xiangxi cattle.

One reasonable way is to use a global-ancestry-inference algorithm to determine the proportion of ancestry and then use these ancestry groups as references for those individuals. Based on the results of ADMIXTURE and Admixtools2, we selected Chinese indicine, Indian indicine and Hanwoo as reference panels. As expected, most of the local Loudi cattle genome fragments were inferred to be Chinese indicine-like, mainly distributed on chromosomes 7, 11, 13 and 19 (Figure 3B). A total of 146 genes were annotated in the 290 Chinese indicine segments of Loudi cattle (Appendix A). The enrichment analysis revealed that the most enriched pathway was ‘glutamatergic synapse’ (GO:0098978, *p*-value = 1.19 × 10^−5^). Many of the pathways that were significantly enriched (*p*-values < 0.05) were related to environmental adaptation and homeostasis, such as ‘phosphonate and phosphinate metabolism’ (bta00440, *p*-value = 3.28 × 10^−2^), ‘renin secretion’ (bta04924, *p*-value = 5.51 × 10^−3^) and ‘kidney development’ (GO:0001822, *p*-value = 3.28 × 10^−2^) (Figure 3C, Appendix A). Eighteen Indian fragments mainly annotated genes related to neurological development, such as *POLRMT* [41], *NRG2* [42] and *TCF12* [43].

### 3.4. Patterns of Selection

We obtained 1685 candidate genes by θπ, whereas 758 positively selected genes were detected by CLR (Figure 4A, Appendix A), which overlapped with 214 genes. Many of these genes have been shown to be associated with steady-state, heat tolerance and renal function; for example, *EXOC3* maintains organ homeostasis by driving lipid metabolism [44], *FGR* causes kidney injury [45], *GPR3* is a receptor gene involved in adipose thermogenesis [46] and *MLX* regulates metabolism [47]. To further understand the differential selection signal between Loudi cattle and Xiangxi cattle, *F*_ST_ and XP-EHH tests were performed (Figure 4A, Appendix A). The overlapping region of any two methods was considered a candidate region, and 599 candidate genes were identified for functional enrichment in these candidate regions (Appendix A). Intermediates of the ‘citrate metabolic process’ (GO:0006101, *p*-value = 3.11 × 10^−4^) regulate hypoxia-inducible factors (HIF), the key mediators of adaptation to hypoxia [48]. Other pathways closely related to homeostasis included ‘locomotor rhythm’ (GO:0045475, *p*-value = 4.74 × 10^−5^) and ‘collecting duct acid secretion’ (bta04966, *p*-value = 2.58 × 10^−3^). It is worth noting that five candidate genes (*CHRNE*, *MINK1*, *NR3C1*, *C19H17ORF107* and *ARHGAP26*) were detected via the four methods mentioned above, indicating that they were subjected to strong selection in Loudi cattle (Figure 4B). Moreover, *NR3C1* showed strong positive selection in Loudi cattle (Figure 4C).

We examined the missense mutations in these five genes and found three distinct allelic patterns (Figure 4D): *NR3C1* is a glucocorticoid receptor gene that is closely related to environmental exposure and homeostasis. Three cases of homozygous A-to-G substitution at c.521 of the *NR3C1* gene were found in Loudi cattle but not in Xiangxi cattle. *MINK1* induces phosphorylation of glucocorticoid receptors [49]. Allele C-to-G substitution at c.1615 of the *MINK1* gene was significantly associated with Loudi cattle (*p*-value = 6.27 × 10^−3^, Fisher’s exact test, compared to Xiangxi). Allele A-to-G substitution at c.1615 of the *C19H17orf107* gene was also significantly associated with Loudi cattle (*p*-value = 3 × 10^−2^, Fisher’s exact test, compared to Xiangxi).

## 4. Discussion

Genetic variation reflects the historical selection and evolutionary pressures experienced as these cattle breeds developed. Here, we conducted a whole-genome sequence-based study of the genomic diversity, ancestral inference and selective signatures in Loudi cattle. The ancestral contributions of Loudi cattle came from East Asian taurine, Chinese indicine, European taurine and Indian indicine (Figure 2C). The LD decay pattern of Loudi cattle was similar to that of Xiangxi cattle, confirming the high genetic diversity of Loudi cattle (Figure 1C,D). As indicated by the NJ tree results, two of the three bulls had significant Indian indicine ancestry, which may have further affected the offspring lineage proportion of the Loudi cattle (Figure 2B).

Our results indicate that Loudi cattle and Xiangxi cattle had the same ancestral lineage components. Loudi cattle retained more indicine lineage, while Xiangxi cattle were more influenced by taurine cattle. Xiangxi cattle have become a famous breed after thousands of years of optimization, and the admixture event is after the earliest evidence for indicine cattle in China (~3000 years ago) [50]. These findings are supported by a qpGraph analysis, and the admixture event of Xiangxi cattle occurred at the intermediate node of the Chinese indicine, while the admixture event of Loudi cattle was later than that of Xiangxi cattle and occurred in recent times (Figure 3A). We speculate that the *Bos indicus* lineage of Loudi cattle is influenced by a special cattle breed with both early and recent Chinese indicine lineages. It is important to note that our samples are all modern domestic cattle and span a large geographical area, so the presumed gene flow and admixture events may fall into local optima at the genome level rather than being fully consistent with historical events.

The allele introgression between species can influence the evolutionary and ecological fate of species exposed to novel environments [51]. Of the three genes we focused on, the *NR3C1* gene appeared in the Chinese fragment inferred by RFmix analysis (Appendix A), indicating that the A-to-G substitution at c.521 was spread from Chinese indicine cattle to Loudi and Xiangxi cattle. The humid and hot environment made the mutation burden of Loudi cattle more obvious. The allele C-to-G substitution at c.1615 of the *MINK1* gene belongs to the difference between indicine cattle and taurine cattle and may be one of the advantageous alleles for immune, sensory and heat adaptation [52,53]. Allele A-to-G substitution at c.1615 of the *C19H17orf107* gene showed a specific pattern in Loudi cattle; however, the function of the *C19H17orf107* gene was unclear.

Currently, extreme weather is becoming more severe and is bound to have a huge impact on the survival and production of cattle [53,54]. Of all the candidate regions in Loudi cattle, we observed some genes associated with homeostasis, which may be useful in coping with climate extremes. The genes *GLS* and *GLS2*, involved in reactive oxygen species metabolic processes, were enriched in pathways ‘proximal tubule bicarbonate reclamation’ and ‘glutamatergic synapse’, indicating that Loudi cattle may have evolved stronger renal function to cope with the humid and hot environment. It is more strategically important than ever to preserve as much of the cattle diversity as possible, to ensure a prompt and proper response to the needs of the future environment.

## 5. Conclusions

This study provides a theoretical basis for analyzing the genetic mechanism of Loudi cattle with excellent environmental adaptation and homeostasis, which also lays a foundation for the genetic breeding work of Loudi cattle in the future.

## Figures and Tables

**Figure 1 biology-11-01775-f001:**
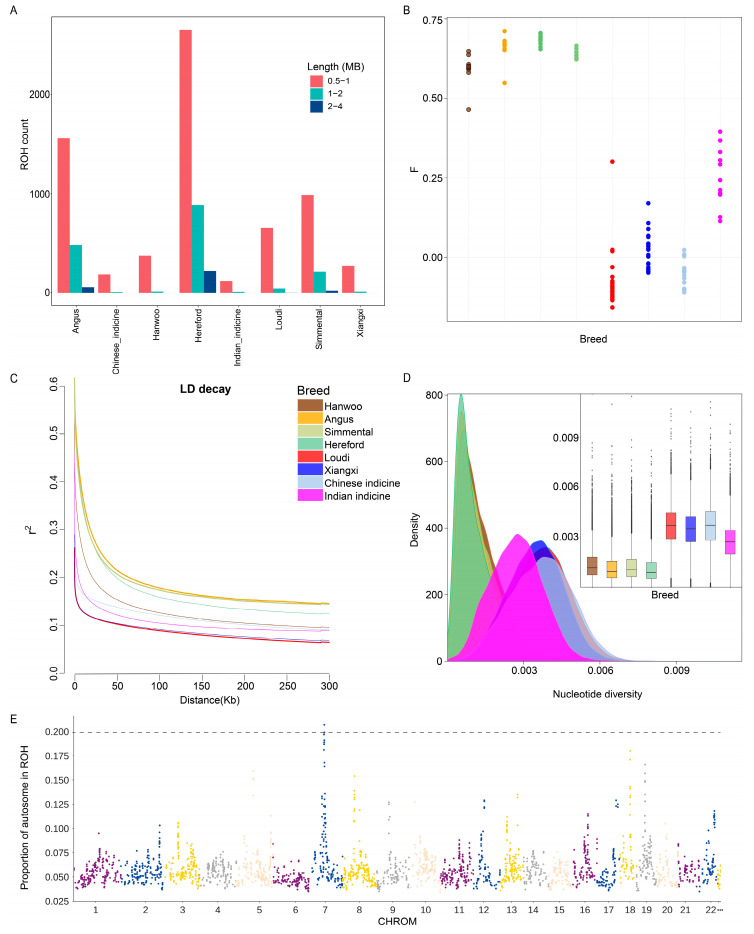
Genetic diversity analysis of 125 samples from 8 populations, among which (**B**–**D**) shared the same legend and color. (**A**) Bar chart of ROH analysis. Red represents ancient inbreeding, while green and blue represent historical and recent times, respectively. (**B**) F-coefficient for genomic heterozygosity estimates. (**C**) Decay of LD in the 8 cattle populations, with one line per population. (**D**) The nucleotide diversity for each group by density plots and box plots. (**E**) Manhattan plots of ROH frequencies in Loudi cattle.

**Figure 2 biology-11-01775-f002:**
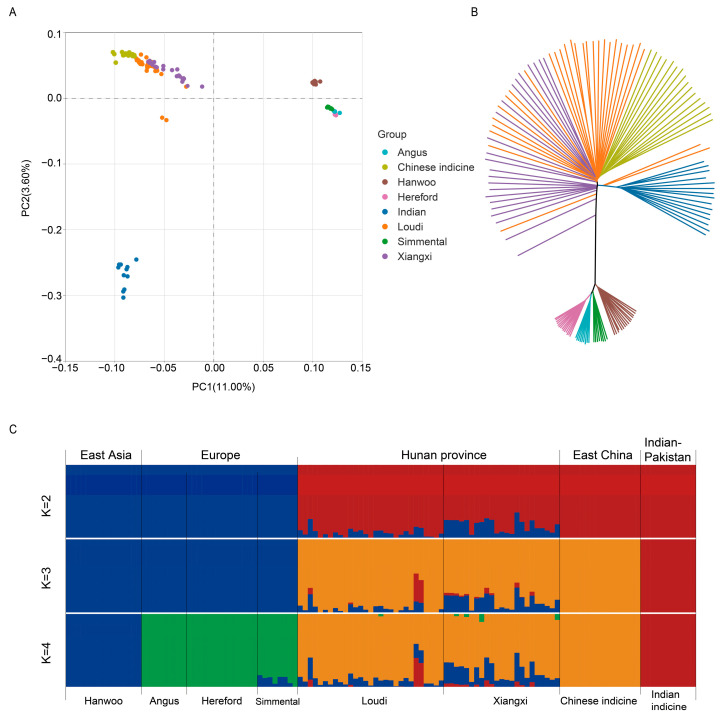
The structures of 125 samples from 8 populations. (**A**) PCA. Principal components 1 (11.00%) and 2 (3.60%) for the 125 cattle. (**B**) Phylogenetic tree. Phylogenetic relationships were estimated using the neighbor-joining method. (**C**) Genetic structure of cattle using ADMIXTURE when K ranged from 2 to 4. When K = 4, the CV error is the smallest.

**Figure 3 biology-11-01775-f003:**
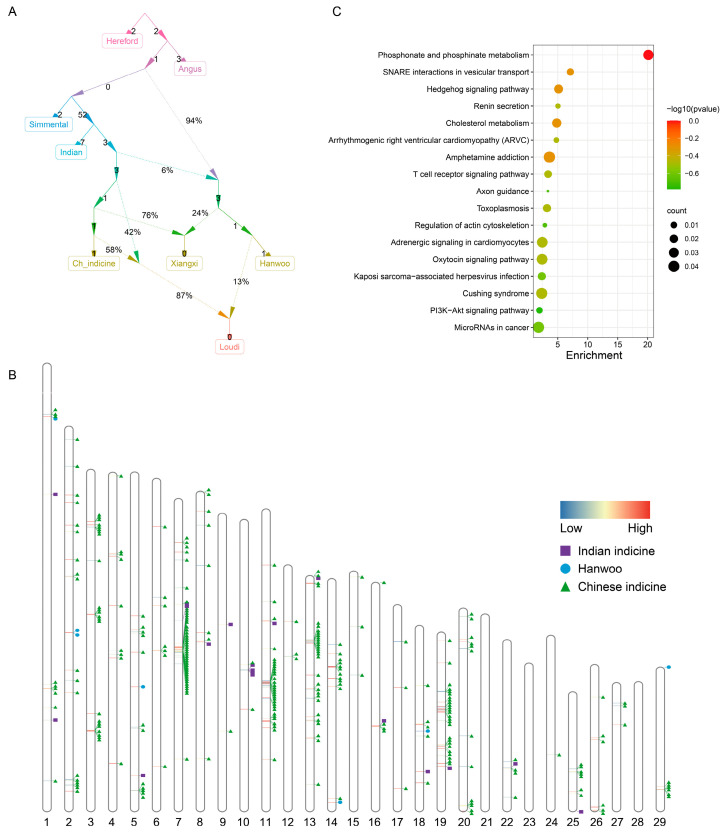
Admixture events and ancestry inference in Loudi cattle. (**A**) Iterative building of admixture graphs using Admixtools2-qpGraph. (**B**) Identification of the local segments in which proportions of a certain ancestry were significantly higher than the proportion in the whole genome in Loudi cattle. (**C**) The KEGG pathways from the enrichment analysis of genes with excessive Chinese indicine proportions.

**Figure 4 biology-11-01775-f004:**
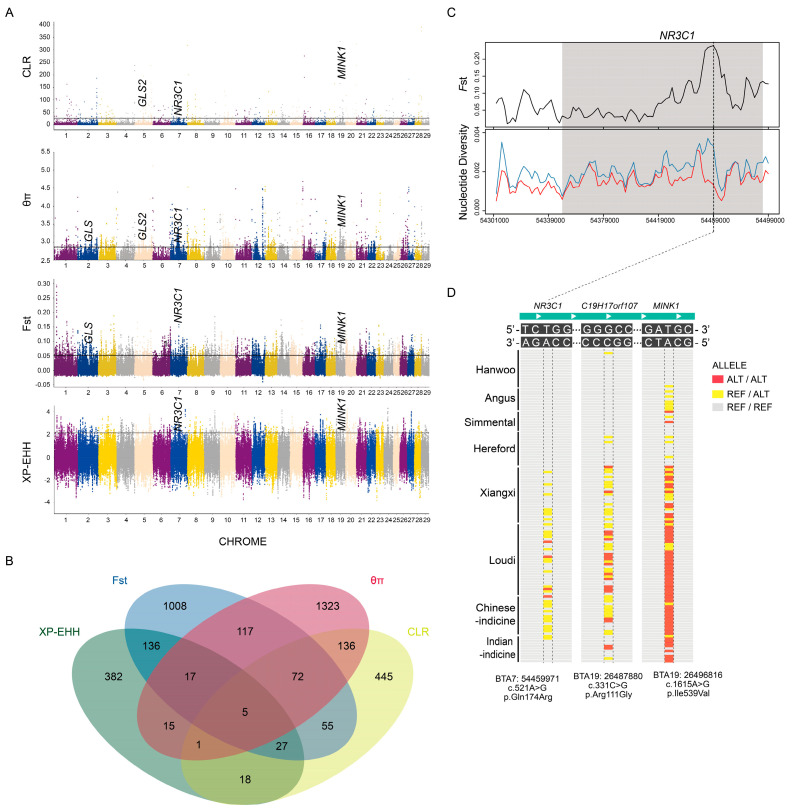
Selective sweep analysis of Loudi cattle. (**A**) Manhattan plot showing the four methods of CLR, θπ, *F*_ST_ and XP-EHH. (**B**) Venn diagram showing the gene overlaps among θπ, CLR, *F*_ST_ and XP-EHH. (**C**) *F*_ST_ and nucleotide diversity plots in the *NR3C1* gene region. The red line represents the reference group of Xiangxi cattle. (**D**) Regional highlight of the missense mutation of *NR3C1*, *C19H17orf107* and *MINK1*.

## Data Availability

The datasets presented in this study can be found in online repositories. The names of the repository/repositories and accession numbers can be found in the article/Appendix A.

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
