# Peer review of "Genomics, Origin and Selection Signals of Loudi Cattle in Central Hunan"

_biology, 2022, doi:10.3390/biology11121775_

Round 1
Reviewer 1 Report
The authors investigated the Loudi cattle, regarding their admixture level, ROH distribution, population structure, and genomic regions showing selection signs based on NGS data generated in this study and on sequence data deposited by other researchers.
The applied techniques adequately describe the breed, give insight into its history and highlight genomic regions worth studying in more detail.
I suggest the manuscript for publication after minor modifications.
Notices:
Line 17: '...world cattle...' is a kind of exaggeration. Maybe it is better to use 'carefully selected cattle breeds' instead.
Line 19: The entire sentence is hanging there. Its location might be better after the following sentence. Or it is just not necessary here.
Line 49-50: The wisdom of any proverb is undeniable, but I do not feel it belongs here. Please delete.
Line 49-50: The entire paragraph, especially 'Chinese civilisation has changed...' lacks a reference. Please find one.
Please refer to the Figures in the Discussion part, as well.
Author Response
Response to Reviewer 1 Comments
Point 1: Line 17: '...world cattle...' is a kind of exaggeration. Maybe it is better to use 'carefully selected cattle breeds' instead.
Response 1: Modifications (Line 18) have been made in the re-submit manuscript.
Point 2: Line 19: The entire sentence is hanging there. Its location might be better after the following sentence. Or it is just not necessary here.
Response 2: We made some changes (Line 19) to make the sentences flow more smoothly.
Point 3: Line 49-50: The wisdom of any proverb is undeniable, but I do not feel it belongs here. Please delete.
Response 3: The proverb has been removed from in the re-submit manuscript (Line 55).
Point 4: Line 49-50: The entire paragraph, especially 'Chinese civilisation has changed...' lacks a reference. Please find one.
Response 4: This paragraph mainly refers to Loudi local newspapers and factbook. Quotations and sources (in Chinese) have been added in the resubmitted manuscript.
Point 5: Please refer to the Figures in the Discussion part, as well.
Response 5: Modifications have been made in the re-submit manuscript.

Reviewer 2 Report
The manuscript is based on WGS data for Hunan local cattle breed.
Comments
The strategy for collecting samples from Loudi cattle breed should be clearly stated in the Materials and Methods.
Were samples collected from a single or multiple farms?
No data are available whether the selected animals were checked for relatedness. However, this is relevant to provide unbiased results and their interpretation.
L21-22 The authors declare that «Loudi cattle and Xiangxi cattle are very similar in exterior conformation, which brings great trouble to local cattlemen».
This phrase should be treated carefully.
Did these breeds originate from the common ancestor?
What is demographic history of the breeds?
Were the breeds intercrossed?
Are the authors sure that they had sampled from the purebred Loudi cattle and no Xiangxi cattle?
L 283 -284 «Our results indicate that Loudi cattle and Xiangxi cattle had evolved from the same ancestral population» However, the results showed in the Figures 2F-C (especially in Admixture plot) do not provide evidence that Loudi cattle is differentiated from Xiangxi breed while the admixed events are clear.
I recommend revising thoroughly the results presented in Figure 2.
The photographs of Loudi cattle and Xiangxi cattle will attract attention of potential readers.
Besides I recommend finding ROH islands to check the overlapped genes.
Author Response
Response to Reviewer 2 Comments
Point 1: The strategy for collecting samples from Loudi cattle breed should be clearly stated in the Materials and Methods.
Were samples collected from a single or multiple farms?
No data are available whether the selected animals were checked for relatedness. However, this is relevant to provide unbiased results and their interpretation.
Response 1: Related Description have been added to the methods in the resubmit manuscript. Due to the lack of local cattle reservation areas in Loudi, we selected 29 unrelated cattle from multiple farms in Loudi area.
Point 2: L21-22 The authors declare that «Loudi cattle and Xiangxi cattle are very similar in exterior conformation, which brings great trouble to local cattlemen».
This phrase should be treated carefully.
Response 2: Modifications have been made in the re-submit manuscript.
Point 2: Did these breeds originate from the common ancestor?
What is demographic history of the breeds?
Were the breeds intercrossed?
Are the authors sure that they had sampled from the purebred Loudi cattle and no Xiangxi cattle?
Response 3: Hunan Province, located in central China. The demographic history and origins of this area have been studied in previous work (https://www.nature.com/articles/s41467-018-04737-0#rightslink). They are predominantly of Chinese indicine ancestry, and Indian indicine cattle and East Asian taurine cattle introgression into them at different times. Therefore, these breeds are hybrids and originate from the common ancestor (Chinese indicine cattle), which is also validated by our analysis. In addition, the Xiangxi cattle was sampled from single conservation farm in western Hunan Province, ~500 kilometers away from Loudi area. After confirmation by the farmer, all the cattle we sampled were purebred Loudi cattle and had no relatives.
Point 4: L 283 -284 «Our results indicate that Loudi cattle and Xiangxi cattle had evolved from the same ancestral population» However, the results showed in the Figures 2F-C (especially in Admixture plot) do not provide evidence that Loudi cattle is differentiated from Xiangxi breed while the admixed events are clear.
I recommend revising thoroughly the results presented in Figure 2.
The photographs of Loudi cattle and Xiangxi cattle will attract attention of potential readers.
Besides I recommend finding ROH islands to check the overlapped genes.
Response 3: More accurate sentence was changed in the re-submit manuscript (line 322): “Loudi cattle and Xiangxi cattle had the same ancestral lineage components”.
We tried to visualize K=5 in Admixture analysis , and although Xiangxi formed a new color, the color representing the origin of other ancestors was covered, and the optimal number of clusters was 4 according to the minimum CV error rate.
Attached Figure 2C when K=5 (we're not sure if it needs to be replaced):
In addition, we added ROH islands analysis (line 135-138; line 214-218). Although ROH islands with frequencies of 20% are rare, fortunately we annotated some genes related to heat resistance and anti-inflammation, and our final gene NR3C1 on chromosome 7 is also shown in the Figure 1E (~12%). Thanks for your suggestion.

Reviewer 3 Report
Title: Genomics, origin and selection signals of Loudi cattle in central 2 Hunan
Keywords: Whole-genome resequencing; Genetic diversity; Origin; Selection signals; Loudi cattle
There are repeated keywords in the title. Authors must choose only one location. You must not repeat them.
In figure 1 B, need to enter the X axis title;
In figure 1 C, it is not possible to identify the breeds by the colors shown in the graph.
Insert the axes for the box plot in figure D
Perhaps it would be interesting to unlock figure 1, turning it into 4 figures.
It would improve the view quality.
It would be interesting to put an image of the cattle of the breeds. If possible from the animals collected. { Loudi cattle; Chinese cattle; and Xiangxi cattle}
Author Response
Response to Reviewer 3 Comments
Point 1: There are repeated keywords in the title. Authors must choose only one location. You must not repeat them.
Response 1: I'm not sure which keywords are repeated in the title.
Point 2:
In figure 1 B, need to enter the X axis title;
In figure 1 C, it is not possible to identify the breeds by the colors shown in the graph.
Insert the axes for the box plot in figure D
Perhaps it would be interesting to unlock figure 1, turning it into 4 figures.
It would improve the view quality.
It would be interesting to put an image of the cattle of the breeds. If possible from the animals collected. { Loudi cattle; Chinese cattle; and Xiangxi cattle}
Response 2: Thanks for your suggestion. The Figure 1 have been adjusted in the resubmitted manuscript, but it is difficult to split them further due to the large number of pictures (We want them in the text and not in the supplementary material). We also added instructions (line 220): Figure 1B-D shared legend and color.

Reviewer 4 Report
Authors of the manuscript describes the study of the origin and selection signals of Loudi cattle. The study of the genetic diversity of local breeds usually well adapted to the harsh climate environment is an interesting and important task. In this regard, I believe that this work provides a theoretical platform for a better understanding of the complex genomic architecture picture of the studied breeds. At the same time, there are a number of questions to the authors:
- lines 173-177. Investigating of ROH patterns. Please add a mention of Figure 1A to the text.
- lines 176-177 and lines 215-217
On the one hand (lines 176-177) Loudi cattle exhibited a heavier ROH burden than Xiangxi cattle, suggesting a longer history of breeding. On the other hand (lines 215-217) the topological map also supported the result that the infiltration of taurine cattle was more serious in Xiangxi cattle (~24%), and the admixture event of Loudi cattle was much later than that of Xiangxi cattle.
Does this mean that Loudi cattle have been isolated from contact with taurine cattle for a long history of breeding while Xiangxi has not?
Is Xiangxi breed an offshoot of the Loudi breed? If so, did this branch separate due to hybridization with taurine cattle or due to other factors?
What is the number of the Loudi and Xiangxi livestock? Are these cattle breeds the rare local breeds or are they widespread?
A brief history of the creation of these breeds will help to better understand interbreed relationships. Could the authors add a few words about the breeds history to the text?
- line 199. Is the Figure 3C should be Figure 2C?
What is the optimal number of clusters in the Admixture analysis? Will Loudi and Xiangxi separate from each other if K will more than 4?
Figure 4A shows the genes (MINK1, NR3C1, GLS and GLS2) identified by at least two methods. Three genes (C19H17orf107, CHRNE and ARHGAP26) detected by four methods absent in the Figure4A. Why?
Author Response
Response to Reviewer 4 Comments
Point 1: - lines 173-177. Investigating of ROH patterns. Please add a mention of Figure 1A to the text.
Response 1: Modifications (Line 206) have been made in the re-submit manuscript.
Point 2: - lines 176-177 and lines 215-217
On the one hand (lines 176-177) Loudi cattle exhibited a heavier ROH burden than Xiangxi cattle, suggesting a longer history of breeding. On the other hand (lines 215-217) the topological map also supported the result that the infiltration of taurine cattle was more serious in Xiangxi cattle (~24%), and the admixture event of Loudi cattle was much later than that of Xiangxi cattle.
Does this mean that Loudi cattle have been isolated from contact with taurine cattle for a long history of breeding while Xiangxi has not?
Is Xiangxi breed an offshoot of the Loudi breed? If so, did this branch separate due to hybridization with taurine cattle or due to other factors?
What is the number of the Loudi and Xiangxi livestock? Are these cattle breeds the rare local breeds or are they widespread?
A brief history of the creation of these breeds will help to better understand interbreed relationships. Could the authors add a few words about the breeds history to the text?
Response 2: Some notes (Line 71-74) have been added to the resubmitted manuscript. As explained on lines 68-74, Loudi cattle have been isolated from contact with taurine cattle for a long history of breeding due to customs and geographical environment. Nowadays, the breeding system of Xiangxi cattle is relatively mature, while the number of cattle in Loudi is relatively small and there is no large-scale breeding farm (We have added the strategy for collecting samples from Loudi cattle in the Materials and Methods of the resubmitted manuscript).
We consider Loudi cattle and Xiangxi cattle are local cattle breeds (mainly of East Asian indicine lineage), and East Asian taurine cattle and Indian indicine cattle introgression into them at different times. The demographic history and origins of major regions of the world have been reported in previous work (https://www.nature.com/articles/s41467-018-04737-0#rightslink).
Point 3: - line 199. Is the Figure 3C should be Figure 2C?
What is the optimal number of clusters in the Admixture analysis? Will Loudi and Xiangxi separate from each other if K will more than 4?
Response 3: Modifications (line 237) have been made in the re-submit manuscript.
We tried to visualize K=5 in Admixture analysis , and although Xiangxi formed a new color, the color representing the origin of other ancestors was covered, and the optimal number of clusters was 4 according to the minimum CV error rate.
Attached Figure 2C when K=5:
Point 4: Figure 4A shows the genes (MINK1, NR3C1, GLS and GLS2) identified by at least two methods. Three genes (C19H17orf107, CHRNE and ARHGAP26) detected by four methods absent in the Figure4A. Why?
Response 4: We've only tagged genes of interest in the Figure 4A. About the other three genes: Almost nothing pertaining to C19H17orf107 can be retrieved; Both CHRNE and ARHGAP26 appear to be associated with neural signaling functions and no significantly different missense mutation sites were found.

Round 2
Reviewer 2 Report
The manuscript was improved.
Author Response
Thank you for your comments.